# The NLRP3 Inflammasome and Its Role in the Pathogenicity of Leukemia

**DOI:** 10.3390/ijms22031271

**Published:** 2021-01-28

**Authors:** Laura Urwanisch, Michela Luciano, Jutta Horejs-Hoeck

**Affiliations:** 1Department of Biosciences, University of Salzburg, 5020 Salzburg, Austria; laura.urwanisch@sbg.ac.at (L.U.); michela.luciano@sbg.ac.at (M.L.); 2Cancer Cluster Salzburg (CCS), 5020 Salzburg, Austria

**Keywords:** NLRP3 inflammasome, IL-1β, autophagy, leukemia

## Abstract

Chronic inflammation contributes to the development and progression of various tumors. Especially where the inflammation is mediated by cells of the innate immune system, the NLRP3 inflammasome plays an important role, as it senses and responds to a variety of exogenous and endogenous pathogen-associated molecular patterns (PAMPs) and damage-associated molecular patterns (DAMPs). The NLRP3 inflammasome is responsible for the maturation and secretion of the proinflammatory cytokines interleukin-1β (IL-1β) and IL-18 and for the induction of a type of inflammatory cell death known as pyroptosis. Overactivation of the NLRP3 inflammasome can be a driver of various diseases. Since leukemia is known to be an inflammation-driven cancer and IL-1β is produced in elevated levels by leukemic cells, research on NLRP3 in the context of leukemia has increased in recent years. In this review, we summarize the current knowledge on leukemia-promoting inflammation and, in particular, the role of the NLRP3 inflammasome in different types of leukemia. Furthermore, we examine a connection between NLRP3, autophagy and leukemia.

## 1. Introduction

Leukemia is a broad group of clonal hematological malignancies affecting the maturation and/or proliferation of cells of myeloid or lymphoid lineages and can be further subdivided into acute and chronic forms. In adults, the most common types of leukemia are acute myeloid leukemia (AML) [1,2], chronic lymphocytic leukemia (CLL) [3,4] and chronic myeloid leukemia (CML) [5], whereas acute lymphocytic leukemia (ALL) [6] occurs mainly in children. These four main types of leukemia differ in the speed of disease development and progression, the type of hematopoietic cells that are affected, the genetic alterations involved and treatment options and outlooks [1,2,3,4,5,6]. Leukemia is the 10th most common cancer in the United States, and although treatment options have improved over the past years, the five-year relative survival rate is comparatively low at 63.7% (2010–2016), and leukemia is still listed as the seventh leading cause of cancer death in the United States [7].

According to the concept of immune surveillance of cancer, the immune system is usually able to recognize and eliminate transformed cells [8]. However, tumor cells often develop mechanisms to evade detection and destruction by the immune system, and inflammation can even support cancer development and progression. Especially, chronic inflammation drives many types of cancers by promoting mutagenesis, preventing tumor surveillance, supporting clonal evolution and facilitating tumor spreading, among other effects [9,10]. Therefore, both the avoidance of immune destruction and tumor-promoting inflammation were defined, among others, as hallmarks of cancer by Hanahan and Weinberg [11]. It is well-known that inflammation in the tumor microenvironment is associated with the release of various growth factors and proinflammatory cytokines, such as interleukin-1 (IL-1), IL-4, IL-6, tumor necrosis factor-α (TNF-α), transforming growth factor-β (TGF-β) and IL-10, able to promote tumorigenesis [11,12,13]. This is the case not only for solid tumors but, also, for hematopoietic malignancies such as leukemia and myelodysplastic syndrome (MDS), which are often characterized by strong chronic inflammation stimulated by the overproduction of inflammatory cytokines [14].

Disorders of the hematopoietic system arise from alterations in the proliferation and differentiation of hematopoietic stem cells (HSCs), which are responsible for the production and maintenance of the immune system. Chronic inflammation can cause dysfunctional HSC maturation and the abnormal differentiation of immune cells; thus, malignancies of the hematopoietic system often display an elevated production of proinflammatory cytokines, which is often a predictor of a poor prognosis [15]. In particular, IL-1 has been recognized as a major mediator connecting inflammation and tumor promotion [16]. Among the IL-1 family, which comprises 11 cytokines/ligands and 10 related receptors [17,18], interleukin-1β (IL-1β) stands out as one of the most potent proinflammatory cytokines, initiating and amplifying inflammatory responses [19] and linking the innate and adaptive immune systems. However, when released within the tumor microenvironment or during chronic inflammation, IL-1β can support tumor development and progression by interfering with different mechanisms, as described in detail by Bent et al. [20].

The functional role of IL-1β in hematological malignancies has been elucidated in recent studies and is summarized in several reviews [19,21,22]. Already, in 1989, a study suggested that IL-1 may act as an autocrine growth factor for AML cells [23]. This has recently been confirmed in AML patients, who often exhibit enhanced levels of IL-1β and IL-1 receptors. In the majority of AML patients, IL-1 secreted into the bone marrow microenvironment plays a major role in favoring the clonogenicity of myeloid progenitor cells while preventing the growth of normal precursors [24]. IL-1β is thought to increase cell proliferation by stimulating the production of other growth factors and cytokines, such as granulocyte-macrophage colony-stimulating factor (GM-CSF) [19,24]. Additionally, in ALL patients, a highly dysregulated inflammatory state can be detected, which is characterized by elevated circulating levels of proinflammatory cytokines such as IL-1β, TNF-α and IL-6 [25]. Hematopoietic leukemic cells from the bone marrow of B-cell acute lymphocytic leukemia (B-ALL) are involved in the production of proinflammatory mediators and growth factors in ALL patients. This proinflammatory milieu was shown to stimulate the proliferation and differentiation of normal stem and progenitor cells and to have a long-term adverse effect on normal hematopoietic differentiation fates in the bone marrow [26]. Furthermore, upregulated IL-1 signaling plays an important role in CML by promoting the proliferation and survival of primitive CML stem cells. Thus, IL-1 signaling might be a potential therapeutic target to efficiently kill leukemic stem cells [27]. Indeed, blocking IL-1 signaling with monoclonal antibodies targeting the interleukin 1 receptor accessory protein (IL1RAP and IL1R3) has been shown to have an antileukemic effect on CML and AML cells in vivo [27,28]. Regarding CLL, specific polymorphisms in genes coding for IL-1β and IL-6 have been linked to an increased risk of developing CLL [29].

A crucial mechanism driving inflammatory processes and the production of the inflammatory cytokine IL-1β in immune cells is the activation of the NLRP3 inflammasome, which is the best-characterized member of the inflammasome family. The NLRP3 inflammasome is a cytosolic pattern recognition receptor (PRR) that responds to different pathogen-associated molecular patterns (PAMPs), damage-associated molecular patterns (DAMPs) and metabolic changes. The activation of NLRP3 and formation of the NLRP3 inflammasome leads to caspase-1-mediated maturation and secretion of the proinflammatory cytokines IL-1β and IL-18. In addition, gasdermin-D (GSDMD) cleavage by caspase-1 leads to pore formation in the cell membrane, which results in the inflammatory cell death known as pyroptosis [30]. It is already well-established that inappropriate activation of the NLRP3 inflammasome contributes to the onset and progression of various diseases, including inflammatory disorders (inflammatory bowel diseases and rheumatoid arthritis); neurodegenerative diseases (Parkinson’s disease, Alzheimer’s disease and multiple sclerosis) and metabolic disorders (type 2 diabetes and atherosclerosis) [31]. However, NLRP3 inflammasome formation has also been shown to be a key event in tumorigenesis. Depending on the type of cancer, NLRP3 can have opposing functions, either promoting tumor formation or, as some studies show, counteracting tumor development. This controversial role of NLRP3 in the tumor development of various cancer types was well-summarized by Hamarsheh and Zeiser [32].

Until recently, little was known about the influence of the NLRP3 inflammasome on hematopoietic malignancies; however, the importance of the NLRP3 inflammasome is becoming increasingly evident in hematological diseases. Therefore, this review discusses chronic inflammation and the NLRP3 inflammasome in the context of leukemia and its preforms and briefly summarizes the current knowledge about their interrelationships.

## 2. The NLRP3 Inflammasome

Inflammasomes are multiprotein complexes and an essential part of the innate immune system. They belong to the family of pattern-recognition receptors (PRRs) and link critical microbial and/or endogenous danger signals to caspase-1 activation and the subsequent IL-1β secretion [30,33]. The inflammasome family consists of at least five members: NLRP1, NLRP3, NLRC4, AIM2 and Pyrin. The NLRP3 inflammasome is the best-characterized member of the NLRP subfamily of NOD-like receptors [34,35]. The NLRP3 inflammasome is expressed in innate immune cells (monocytes, macrophages, granulocytes and dendritic cells) but, also, in T and B lymphocytes and hematopoietic stem progenitor cells (HSPCs) [36], where it acts as a sensor of changes in the microenvironment, cell activation and metabolic activity. It consists of three proteins that have to be assembled to form the active complex. These include NOD-like receptor protein 3 (NLRP3), apoptosis-associated speck-like protein (ASC) and pro-caspase-1 [33,37].

NLRP3 itself is a tripartite protein that consists of an amino-terminal pyrin domain (PYD), a central nucleotide-binding and oligomerization domain (NACHT) and a carboxy-terminal leucine-rich repeat (LRR) domain. Upon activation, the PYD domain interacts with the amino-terminal PYD domain of ASC [38]. Whereas the NACHT domain has the ATPase activity necessary for oligomerization of the inflammasome [39], the LRR domain seems to be less important for the activation of NLRP3 [40]. Together with the adaptor ASC—which consists of two protein interaction domains, an amino-terminal PYD domain and a carboxy-terminal caspase recruitment domain (CARD)—and pro-caspase-1 [41], NLRP3 forms the inflammasome complex. The effector pro-caspase-1 consists of an amino-terminal CARD domain, a central, large catalytic domain (p20) and a carboxy-terminal small catalytic domain (p10) [42]. In addition, the recently described NIMA-related kinase 7 (NEK7), a serine-threonine kinase known to be involved in mitosis, was identified as a core component of inflammasome activation [43,44,45,46].

Since the uncontrolled formation of an inflammasome is potentially highly inflammatory, several different signals must interact to ensure that its activation is tightly regulated. Therefore, canonical NLRP3 inflammasome activation is often a two-step process consisting of priming followed by activation [30,37,47].

The priming signal aims to upregulate the expression of inflammasome components such as NLRP3 and pro-IL-1β at the transcriptional level. This occurs when distinct PAMPs and DAMPs are recognized by PRRs such as the membrane-bound toll-like receptors (TLR) or the cytoplasmic nucleotide-binding oligomerization domain-containing protein (NLR), such as, for example, NOD1/2. Additionally, the recognition of endogenous cytokines such as IL-1β and TNF by their receptors induces nuclear factor-κB (NF-κB), which activates the transcription of inflammasome components and primes the cell for inflammasome activation [48,49].

After the priming step, the inflammasome can be activated by various stimuli, as recently summarized by Swanson et al. [30]. These usually do not act directly on NLRP3 but induce cellular stress and intracellular events that are then sensed by NLRP3, such as K^+^ efflux [50,51,52,53], Ca^2+^ flux [54,55], Cl^−^ efflux [56], mitochondrial dysfunction and reactive oxygen species (ROS) production [57,58,59] or lysosomal damage [60,61]. As a result of these processes, the inflammasome can be activated, which allows proximity-induced autoproteolytic cleavage of pro-caspase-1 between p20 and p10 to generate the active caspase-1 tetramer, which is now able to proteolytically cleave the proinflammatory cytokines pro-IL-1β and pro-IL-18 into their biologically active forms [30,37,42,47]. Caspase-1 also cleaves GSDMD, enabling its amino-terminus to form a pore in the cell membrane, thus initiating a proinflammatory form of lytic programmed cell death known as pyroptosis [62] (Figure 1). This type of cell death is characterized by cell swelling, membrane rupture and, subsequently, by the release of inflammatory compounds into the extracellular space, such as IL-1β, IL-6 and IL-18 [63,64].

However, the NLRP3 inflammasome can also be activated via a noncanonical pathway, in which cellular priming is unnecessary for inflammasome activation [30,37,65,66,67] and, in some cell types such as human monocytes, in a one-step process [68,69]. This indicates that the activation does not necessarily have to consist of two steps.

## 3. The Role of the NLRP3 Inflammasome in Different Types of Leukemia

Several studies have shown that dysregulated IL-1β secretion and/or signaling in leukemia, especially AML, ALL and CML, positively correlates with disease progression and poor prognosis. In addition, recent data also indicate that the NLRP3 inflammasome plays an important role in hematological malignancies (Table 1 and Figure 2), as in, for example, myelodysplastic syndrome (MDS), myeloproliferative neoplasms (MPNs) and leukemia [36]. The genetic polymorphisms and expression profiles of NLRP3 and related genes have been determined in MDS, AML, ALL and CML, revealing that certain polymorphisms in IL-1β, IL-18, NF-κB or NLRP3 could be potential predictors of these malignant diseases [70,71,72,73].

### 3.1. Myelodysplastic Syndrome (MDS)

The term MDS encompasses a heterogeneous group of preleukemic HSC malignancies caused by abnormal and ineffective hematopoiesis. MDS bone marrow precursors are characterized by excessive programmed cell death, chromosomal abnormalities and somatic gene mutations, with the tendency to transform into AML [81]. It was suggested that NLRP3 inflammasome activation serves as a driver of the MDS phenotype. In particular, the alarmin S100A9 and/or founder gene mutations lead to the generation of ROS and, consequently, to pyroptosis by activating the NLRP3 inflammasome and β-catenin, thereby ensuring the propagation of MDS clones. By blocking the inflammasome signaling pathway, normal hematopoiesis was effectively restored, highlighting the NLRP3 inflammasome as a potential therapeutic target for MDS patients [74]. Another study supported these findings, showing that S100A9 expression is elevated in MDS patients and promotes the senescence phenotype of bone marrow stromal cells via Toll like receptor 4 (TLR4) signaling, NLRP3 inflammasome formation and IL-1β secretion [76].

### 3.2. Acute Myeloid Leukemia (AML)

In addition to MDS, the NLRP3 inflammasome has also been implicated in the pathogenic phenotype of other hematological diseases. For example, a recent study showed that the oncogenic Kras^G12D^ mutation, which occurs in several types of leukemia, not only promotes cancer development and progression via constantly activated RAS/MEK/ERK signaling (also known as mitogen-activated protein kinases (MAPK) pathway) but, also, by activating the NLRP3 inflammasome, thereby promoting myeloproliferation and cytopenia. This effect is reversible in Kras^G12D^ murine models showing NLRP3 deficiency in the hematopoietic system or by the pharmacological inhibition of NLRP3 inflammasome activation. The pathology stems from Kristen rat sarcoma viral oncogene homolog-Ras-related C3 botulinum toxin substrate 1 (KRAS-RAC1) activation stimulating the production of ROS, a well-known trigger for the activation of the NLRP3 inflammasome. This important role of the NLRP3 inflammasome in the pathogenesis of myeloid malignancies and the newly identified KRAS/RAC1/ROS/NLRP3/IL-1β axis has been demonstrated in chronic myelomonocytic leukemia (CMML), juvenile myelomonocytic leukemia (JNNL) and AML patients harboring the KRAS mutation. These findings highlight the potential central role of the NLRP3 inflammasome in hematological disorders, providing a promising target for therapeutic approaches, especially in KRAS-mutated myeloid malignancies [77].

Regarding AML, there is also evidence that the increased NLRP3 expression in bone marrow mononuclear cells (BMMCs) and peripheral blood mononuclear cells (PBMCs) of newly diagnosed patients correlates with the enhanced expression of the aryl hydrocarbon receptor (AHR). The authors further described an increased population of T-helper 22 (Th22) cells in the peripheral blood of newly diagnosed AML patients, while the Th1 proportion is reduced. Since AHR is involved in the differentiation of Th cell subsets, the authors hypothesized that the NLRP3/AHR axis might be involved in regulating Th cell subset differentiation in AML [78]. While it was previously suggested that AML is characterized by enhanced Th22 and reduced Th1 levels, with Th22 cells being involved in promoting the pathogenesis of leukemia, the underlying mechanisms are barely defined [82]. Thus, further studies are urgently required to confirm this hypothesis.

### 3.3. Acute Lymphocytic Leukemia (ALL)

High NLRP3 inflammasome activity not only promotes carcinogenesis, it also carries the additional risk of causing anticancer-drug resistance, as Paugh et al. showed [79]. ALL is often treated with glucocorticoids, which regulate many physiological processes and alter the transcriptional programs of cells, such that the proliferative capacity of ALL cells is diminished and apoptosis is induced. Thus, patients whose ALL cells are sensitive to glucocorticoids have a significantly better prognosis than those whose cells are resistant to the treatment [83,84,85]. Moreover, ALL cells that are resistant to glucocorticoids have significantly higher expression levels of NLRP3 and caspase-1. Caspase-1 cleaves the glucocorticoid receptor, thereby blunting the effects of the glucocorticoids. The inhibition or knockdown of caspase-1 with short hairpin RNA (shRNA) restored the glucocorticoid sensitivity of caspase-1-overexpressing ALL cells. This suggests that NLRP3 or caspase-1 inhibitors might improve the treatment of ALL patients by reversing the resistance to glucocorticoids [79].

### 3.4. Chronic Lymphocytic Leukemia (CLL)

The studies described so far have all reported that NLRP3 is upregulated in different types of leukemia and has a tumor-promoting effect through different mechanisms. However, there is one study that focused on CLL that stated the opposite. Salaro et al. showed that NLRP3 was significantly downmodulated in CLL lymphocytes compared to those of healthy donors, whereas the P2X7 receptor (P2X7R) was overexpressed [80]. P2X7R is mainly known as an activator of the NLRP3 inflammasome [30,86], but it also prevents apoptosis and promotes cell proliferation [87]. The expression of P2X7R is controlled by NLRP3, such that the downmodulation of NLRP3 drives P2X7R expression and simultaneously contributes to tumor growth, whereas NLRP3 overexpression inhibits cell proliferation and induces cell death. Thus, these findings indicate that NLRP3 may act as a negative regulator of tumor growth in CLL [80].

## 4. The NLRP3 Inflammasome as a Therapeutic Target

Since the NLRP3 inflammasome plays an important role not only in hematological diseases but, also, in many inflammatory diseases [88] and cancers [89,90], it has gained special interest as a promising therapeutic target. Therefore, several pharmacological inhibitors of the NLRP3 inflammasome have been developed to intervene at different levels in the complex signaling pathway, as recently summarized [86,91].

To date, the only drugs in clinical use for NLRP3-related diseases are those targeting IL-1β with IL-1β antagonists or recombinant IL-1β receptor antagonists, such as canakinumab [92,93], anakinra [93] and rilonacept [93,94], which are already approved by the US Food and Drug Administration (FDA) [93]. However, these drugs are not used for the treatment of hematologic diseases, and, in general, there are only limited data on the potential therapeutic use and efficacy of IL-1 inhibitors in hematopoietic disorders. Nevertheless, some studies have already shown that monoclonal antibodies against IL1RAP, the coreceptor of IL-1R1, suppress the proliferation of leukemic stem cells (LSCs) in AML [28,95] and CML models [27]. In addition, in a mouse model of CML, IL-1R antagonists (IL-1Ra) in combination with nilotinib [96], a BCR-ABL tyrosine kinase inhibitor, reduced the number of leukemic cells in blood and bone marrow, as well as the self-renewal potential of LSCs significantly better than nilotinib therapy alone [97]. Additionally, the IL-1Ra anakinra was shown to improve the myeloproliferation and cytopenia phenotypes in Kras^G12D^-mutated leukemia mouse models [77]. Since IL-1β can also be produced by inflammasome-independent pathways or other inflammasomes, specific IL-1β inhibitors may also lead to unintended immunosuppressive effects. Therefore, inhibiting the NLRP3 inflammasome with more specific pharmacological inhibitors might be more beneficial for the treatment of NLRP3-driven diseases. Several direct NLRP3 inhibitors have been discovered. Here, we discuss seven recently identified and promising direct or indirect pharmacological inhibitors of NLRP3 inflammasome activation and their therapeutic potential (Table 2).

The most potent and specific NLRP3 inhibitor is the compound MCC950 (CRID3/CP-456773). It is known to specifically block both canonical and noncanonical NLRP3 activation and IL-1β secretion in mouse and human macrophages in vitro without having an effect on NLRP1, NLRC4 and AIM2 [98,106,107]. MCC950 was reported to interact with the Walker B motif within the NACHT domain of NLRP3 and, thus, block ATP hydrolysis and prevent NLRP3 oligomerization and activation [99,108]. The pharmacological inhibition of NLRP3 inflammasome activation by MCC950 has therapeutic efficacy against various preclinical immunopathological models, such as cryopyrin-associated autoinflammatory syndrome (CAPS), experimental autoimmune encephalomyelitis (EAE) [98], Alzheimer’s disease [106], Parkinson’s disease [109], traumatic brain injury [110], atherosclerosis [111], diabetes [112], steatohepatitis [113] and colitis [114]. In addition, there is evidence that MCC950 also has a therapeutic benefit in a mouse model for Kras^G12D^-mutated myeloid malignancies [77].

Another NLRP3 inhibitor is CY-09, an analog of cystic fibrosis transmembrane conductance regulator (CFTR) channel inhibitor-C172, which was found to effectively and directly inhibit NLRP3 inflammasome activation in vivo in mouse models and ex vivo in human monocytes. CY-09 binds directly to the Walker A motif of the NLRP3 NACHT domain and inhibits its ATPase function and, consequently, NLRP3 oligomerization and activation. Furthermore, CY-09 was shown to have therapeutic effects on mouse models of CAPS and type 2 diabetes (T2D) [100].

OLT1177, a β-sulfonyl nitrile molecule, is another NLRP3 inhibitor that was shown to specifically inhibit both canonical and noncanonical NLRP3 inflammasome activation and, consequently, to reduce caspase-1 activity and IL-1β and IL-18 secretion. OLT1177 showed no effect on the NLRC4 and AIM2 inflammasomes, raising the possibility that it specifically targets the NLRP3 inflammasome. Mechanistically, it was shown that OLT1177 directly binds to NLRP3, blocks its ATPase activity and prevents both the NLRP3–ASC and NLRP3–capase-1 interactions [101]. OLT1177 showed therapeutic benefits in murine models of joint arthritis [115] and multiple sclerosis (MS) [116] and has successfully passed a phase I clinical trial for the treatment of degenerative arthritis and is now being evaluated in a phase II clinical trial [117]. In addition, it is currently undergoing a phase I/II trial for systolic heart failure and Schnitzler’s syndrome (clinicaltrials.gov identifiers NCT03534297 and NCT03595371, respectively).

Tranilast is a tryptophan metabolite analog and was initially recognized as an antiallergic drug and used for the treatment of various inflammatory diseases [118]. Huang et al. identified Tranilast as a specific NLRP3 inhibitor that does not target NLRC4 or AIM2 inflammasomes [102]. It was shown to bind directly to the NACHT domain of NLRP3 and, thus, inhibit the NLRP3–NLRP3 interaction and subsequent oligomerization in an ATPase-independent manner. It was also shown to have therapeutic benefits in gout, CAPS and T2D mouse models [102] and is currently in a phase II clinical trial for CAPS syndrome (clinicaltrials.gov identifier NCT03923140).

Oridonin is the major bioactive component of the plant *Rabdosia rubescens*, which is an over-the-counter herbal medicine that is extensively utilized in traditional Chinese medicine [119] and has been reported to have antitumor, anti-inflammatory and proapoptotic effects [120,121,122]. Oridonin specifically inhibits the NLRP3 inflammasome but not the AIM2 and NLRC4 inflammasomes. It blocks the interaction between NLRP3 and NEK7 by forming an irreversible covalent bond with cysteine 279 of the NLRP3 NACHT domain, which prevents NLRP3 inflammasome assembly and activation [103]. The inhibition of NLRP3 activation by Oridonin has shown both preventive and therapeutic effects in mouse models of T2D, peritonitis and gouty arthritis [103].

Recently, another drug has been discovered that indirectly inhibits NLRP3 further downstream of its signaling cascade. Disulfiram, which has been used for decades in the treatment of chronic alcohol addiction [123], has been identified as an effective inhibitor of GSDMD pore formation by covalently modifying the human/mouse Cys191/Cys192 of GSDMD. Disulfiram does not prevent the processing of IL-1β or GSDMD but exclusively blocks the formation of the pore and, thus, pyroptosis and the release of inflammatory cytokines after activation of the NLRP3 inflammasome [104]. Although its potential to prevent pyroptosis was discovered only recently, it has been known for some time that Disulfiram has antitumor activity in multiple types of tumors, including hematological disorders such as AML [124,125,126,127] and ALL [128].

Another compound that uses a similar mechanism to indirectly inhibit the effects of inflammasome activation is necrosulfonamide (NSA), which has been identified as a chemical inhibitor of GSDMD by binding directly to GSDMD and preventing pyroptosis. This has been demonstrated in sepsis models and suggests that GSDMD inhibitors might also be a potential therapeutic treatment option for NLRP3-related diseases [105].

Although leukemia is an inflammation-driven cancer, the effect of NLRP3 inhibitors on this disease has hardly been investigated yet. However, the previously summarized research findings on this topic suggest that certain types of leukemia may benefit from modifying the NLRP3 inflammasome pathway. Comprehensive future studies are needed to investigate the potential use of NLRP3 inhibitors in NLRP3-driven hematological diseases.

## 5. The NLRP3 Inflammasome and Its Connection to Autophagy

While a plethora of external and host-derived stimuli, as well as oncogenic mutations, have been shown to contribute to NLRP3 activation [30,32,77], autophagy induction may play a role in limiting the inflammasome activity [129,130,131,132]. Autophagy is a “self-eating” process necessary to maintain cellular homeostasis under stress conditions [133]. Studies demonstrating that the loss of the autophagy-related protein Atg16L1 results in increased endotoxin-induced IL-1β production provided the first evidence that autophagy regulates inflammasome activation [129]. Furthermore, it was shown that autophagy may limit IL-1β release by targeting inflammasome components for destruction. This process is mediated by the recruitment of ubiquitinated inflammasome components to the autophagic adaptor protein p62/SQSTM1, which directs ubiquitinated cargos into autophagosomes, leading to their degradation in lysosomes [130]; thus, autophagy may regulate inflammatory responses by eliminating active inflammasomes. NF-κB, which is the main priming factor promoting the expression of NLRP3, was recently shown to also limit NLRP3 activation by inducing the expression of p62/SQSTM1, which, in turn, promotes mitophagy and attenuated IL-1β release [131,132]. Mitophagy is a selective type of autophagy by which damaged mitochondria are removed by Parkin-dependent ubiquitin conjugation and a subsequent recognition by p62/SQSTM1 [134]. The blocking of mitophagy leads to an accumulation of damaged mitochondria, which supports NLRP3 inflammasome activation by the release of ROS [57]. Therefore, the clearance of damaged mitochondria through mitophagy is considered to have a key function in regulating NLRP3 inflammasome activation [131,132,135] and may work as a safety mechanism counteracting the hyperactivation of inflammasomes in chronic inflammation-driven cancer. While these studies indicate that mitophagy inhibits NLRP3 inflammasome activation, the NLRP3 inflammasome conversely can also inhibit autophagy in pathological conditions [136]. This was shown to be due to a mechanism in which the signal molecule TIR-domain-containing adapter-inducing interferon-β (TRIF) is cleaved by caspase-1 [137,138]. Since TRIF, an important adaptor molecule of TLR4 signaling, is an essential part of TLR4-mediated autophagy, the cleavage of TRIF by caspase-1 may decrease the autophagy [139]. Although this inhibitory effect of caspase-1 on autophagy has only been demonstrated in Prion disease [137] and during *Pseudomonas aeruginosa* infection [138], it might also be involved in the pathogenicity of leukemia, which is characterized by high NLRP3 activity and IL-1β secretion. However, further studies are needed to confirm this hypothesis. Recent studies have also suggested that p62/SQSTM1 deficiency compromises cellular homeostasis in leukemia cells through the accumulation of dysfunctional mitochondria and impaired mitochondrial function. In addition, the latter study reports that the deletion of p62/SQSTM1 resulted in a reduced proliferation of leukemia cells and leukemia development in two murine AML models, highlighting that p62 is essential for efficient cell proliferation in AML [140].

The reciprocal inhibition between inflammasome activation and mitophagy/autophagy and the recently highlighted controversial and context-dependent role of mitophagy/autophagy in leukemia point to autophagy modulation as a possible therapeutic target in leukemia [141,142]. Since the proinflammatory environment is known to support tumor cell proliferation and survival, but excessive inflammation can also lead to cell death, unrestricted inflammasome activation might be prevented by the induction of mitophagy in order to dampen NLRP3 inflammasome activation. On the other hand, the suppression of caspase-1-mediated autophagy might lead to increased NLRP3 activation and the increased release of proinflammatory mediators. Under homeostatic conditions, this crosstalk is necessary to prevent excessive inflammation while maintaining the ability to mount an inflammatory response. However, in cancer, this fine balance between NLRP3 activation, inflammation and autophagy might be dysregulated to support cell proliferation, survival and resistance to chemotherapy of malignant cells in an adaptive and context-dependent fashion (Figure 3).

## 6. Conclusions

The NLRP3 inflammasome has become a highly interesting topic in the last several years, and a growing number of studies have focused on the role of NLRP3 in hematopoietic malignancies. Even though NLRP3 is the best-studied member of the inflammasome family, its specific roles in leukemia remain contentious. The functions of the NLRP3 inflammasome in leukemogenesis of the different leukemia types are very distinct; it can both promote and, also, inhibit the emergence and progression of cancer. This appears to depend on several factors, such as the expression level, cancer type, stage of tumorigenesis and certain mutations. Furthermore, the NLRP3 inflammasome seems to be linked to autophagy. This link was recognized previously, but the crosstalk between NLRP3 and autophagy in the context of leukemia is poorly understood. Since the mechanisms behind NLRP3 inflammasome activation and its regulation in leukemia remain controversial, a deeper investigation of the relationship between inflammation, the NLRP3 inflammasome, autophagy and leukemogenesis is needed before this pathway can be effectively targeted by drugs to improve the therapeutic outcomes in leukemia patients.

## Figures and Tables

**Figure 1 ijms-22-01271-f001:**
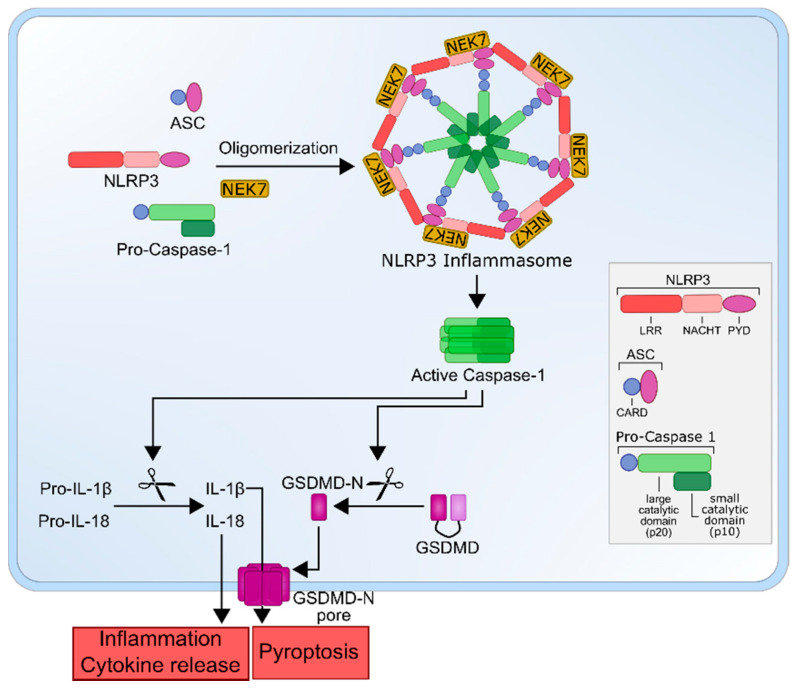
The NLRP3 inflammasome. Oligomerization of NLRP3, ASC and pro-caspase 1 into an active NLRP3 inflammasome leads to caspase-1 activation, which cleaves pro-interleukin-1β (IL-1β) and pro-IL-18 into their active forms, IL-1β and IL-18. Biologically active IL-1β and IL-18 exit the cell and cause inflammation. Additionally, GSDMD is cleaved by capsase-1, whereupon its amino-terminal end forms a transmembrane pore, leading to pyroptosis. NLRP3, NOD-like receptor protein 3; ASC, apoptosis-associated speck like protein; NEK7, NIMA-related kinase 7; GSDMD, Gasdermin-D; GSDMD-N, GSDMD amino-terminal cell-death domain; LRR, leucine-rich repeat; PYD, pyrin domain and CARD, caspase activation and recruitment domain.

**Figure 2 ijms-22-01271-f002:**
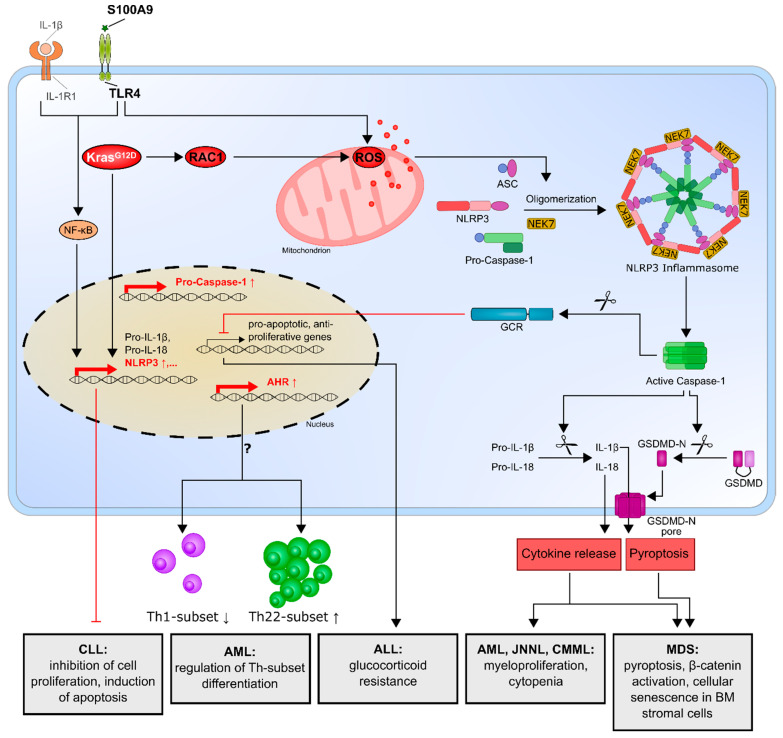
The NLRP3 inflammasome and its effects in leukemia. DAMPs such as alarmin S100A9 induce the transcriptional upregulation of inflammasome components mainly via activation of the transcription factor NF-κB. DAMPs and/or MDS gene mutations lead to the production of ROS, which cause NLRP3 inflammasome activation and, consequently, pyroptotic cell death and altered inflammatory cytokine secretion in MDS patients. In CMML, JNNL and AML, the oncogenic Kras^G12D^ mutation leads to NLRP3/ASC transcription and ROS production via the activation of RAC. Elevated NLRP3 and caspase-1 expression in ALL results in caspase-1-mediated cleavage of the glucocorticoid receptor. Enhanced NLRP3 expression in AML patients correlates with an increased expression of AHR and a shift in Th cell subsets, while NLRP3 overexpression inhibits cell proliferation and induces apoptotic cell death in CLL. ↓ = Activation; ⊥ = Inhibition. NLRP3, NOD-like receptor protein 3; ASC, apoptosis-associated speck-like protein; NEK7, NIMA-related kinase 7; GSDMD, Gasdermin-D; GSDMD-N, GSDMD amino-terminal cell death domain; DAMP, damage-associated molecular pattern; NF-κB, nuclear factor-κB; CLL, chronic lymphocytic leukemia; AML, acute myeloid leukemia; AHR, aryl hydrocarbon receptor; Th subset, T-helper cell subset; ALL, acute lymphocytic leukemia; CMML, chronic myelomonocytic leukemia; JNNL, juvenile myelomonocytic leukemia; RAC, Ras-related C3 botulinum toxin substrate 1; ROS, reactive oxygen species; IL-1β, interleukin-1β; MDS, myelodysplastic syndrome and S100A9, S100 calcium-binding protein A9.

**Figure 3 ijms-22-01271-f003:**
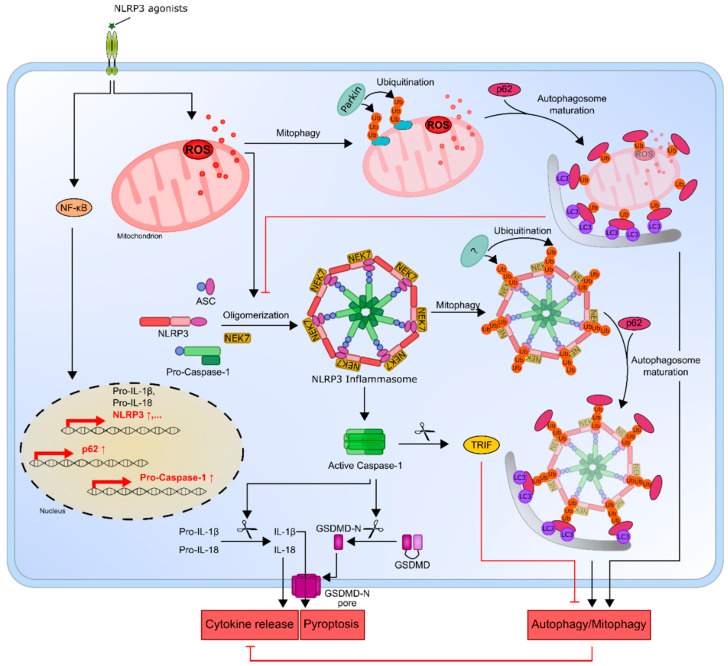
The NLRP3 inflammasome and its connection to mitophagy/autophagy. NLRP3 agonists promote the NF-κB-induced expression of inflammasome components but, also, the expression of p62, which negatively regulates caspase-1 activation by the elimination of mitochondria via mitophagy. NLRP3 activation can also lead to the ubiquitination of NLRP3 inflammasome components and destruction in the autophagosome, thereby limiting IL-1β secretion. In addition, NLRP3 inflammasome activation is able to attenuate autophagy via the caspase-1-mediated cleavage of TRIF, which enhances inflammasome activation. ↓ = Activation; ⊥ = Inhibition. NLRP3, NOD-like receptor protein 3; ASC, apoptosis-associated speck-like protein; NEK7, NIMA-related kinase 7; GSDMD, Gasdermin-D; GSDMD-N, GSDMD amino-terminal cell death domain; NF-κB, nuclear factor-κB; ROS, reactive oxygen species; Ub, ubiquitin; Parkin, E3 ubiquitin protein ligase; LC3, microtubule-associated proteins 1A/1B light chain 3B and TRIF, TIR-domain-containing adapter-inducing interferon-β.

**Table 1 ijms-22-01271-t001:** Key findings of the role of the NLRP3 inflammasome in different hematologic malignancies. NLRP3, NOD-like receptor protein 3; MDS, myelodysplastic syndrome; IL-1β, interleukin-1β; TLR4, Toll-like receptor 4; KRAS, Kristen rat sarcoma viral oncogene homolog; RAC1, Ras-related C3 botulinum toxin substrate 1; ROS, reactive oxygen species; AML, acute myeloid leukemia; CMML, chronic myelomonocytic leukemia; JNNL, juvenile myelomonocytic leukemia; ALL, acute lymphocytic leukemia; and CLL, chronic lymphocytic leukemia.

Type of Hematologic Malignancy	Key Findings	Reference
MDS	NLRP3 inflammasome activation in MDS disorders is responsible for the key biological features of MDS, which drive pyroptotic cell death and β-catenin activation.	[74,75]
	Cellular senescence in bone marrow stromal cells from MDS patients is induced by increased S100A9 expression through TLR4, NLRP3 inflammasome activation and IL-1β secretion.	[76]
AML, CMML, JNNL	Oncogenic Kras^G12D^ mutation activates the KRAS/RAC1/ROS/NLRP3/IL-1β axis and promotes myeloproliferation and cytopenia.	[77]
AML	Enhanced NLRP3 expression correlates with an increased aryl hydrocarbon receptor and might influence T-helper cell differentiation.	[78]
ALL	Overexpression of NLRP3 and caspase-1 is responsible for glucocorticoid resistance through the cleavage of the glucocorticoid receptor by caspase-1.	[79]
CLL	NLRP3 negatively regulates the progression of CLL by promoting the expression of P2X7R, while NLRP3 overexpression inhibits cell proliferation and survival.	[80]

**Table 2 ijms-22-01271-t002:** NLRP3 inhibitors and their targets.

Inhibitor	Inhibition Mechanism	Reference
MCC950	Binds Walker B motif of the NLRP3 NACHT domain; NACHT ATPase inhibitor	[98,99]
CY-09	Binds Walker A motif of the NLRP3 NACHT domain; NACHT ATPase inhibitor	[100]
OLT1177	NACHT ATPase inhibitor	[101]
Tranilast	Binds the NLRP3 NACHT domain and inhibits NLRP3–NLRP3 interaction	[102]
Oridonin	Binds irreversibly to NLRP3 Cys279 and inhibits NLRP3–NEK7 interaction	[103]
Disulfiram	Blocks gasdermin D pore formation and inhibits pyroptosis and cytokine release	[104]
Necrosulfonamide (NSA)	Binds to gasdermin D and prevents pyroptosis	[105]

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
