# Peer review of "The NLRP3 Inflammasome and Its Role in the Pathogenicity of Leukemia"

_ijms, 2021, doi:10.3390/ijms22031271_

Round 1
Reviewer 1 Report
The bibliographic review from Urwanisch et al. describes the current knowledge around the NLRP3 inflammasome in several types of leukemia. The review includes an introduction about leukemia and why the inflammatory component is critical in leukemia onset and progression. Next, the authors focus on the importance of NLRP3 inflammasome and summarize the latest research connecting each single type of leukemia with biological processes involving NLRP3. Finally, authors highlight the potential use of NLRP3 as a therapeutic target and how it is connected to autophagy processes.
The review is written in a clear manner and is easy to follow. However, I believe that the review would benefit from a different distribution of the sections included (detailed below). The use of Figures facilitates the comprehension, and I would like to emphasize the effort made in trying to synthesize the information. The use of bibliographic references chosen by the authors is really up to date, with several citations from the last five years. In summary, the review is of good quality, although some points need to be addressed.
Major points
- I believe that there are several introductory sections (1, 2 and 3) that should be reorganized or, even better, grouped into a single section. I think that it is unnecessary to use a whole section (Section 2. Leukemias) to introduce about the different types of leukemia, since scope readership will be already aware of this classification. Indeed, the connection between leukemia and inflammation from section 3 is much more relevant. I would suggest reducing the content of sections 1 and 2, and merging them with section 3, generating a single introductory section, focused on the importance of inflammation in hematological malignancies, such as leukemia.
- In line with the previous point, Table 1 is not necessary. These data are available in several sources and I don’t think that this is the kind of information sought by readers of this review. For instance, an alternative table including the components and functions (and their role in cancer or hematological diseases) of NLRP3 inflammasome would be much more appropriate for this type of review.
- As I understand from Figure 1, proteins involved in NLRP3 inflammasome formation consist in distinct domains, that need to interact with each other. The domains that constitute each protein and the formation of the complex should be further explained in the NLRP3 Inflammasome section (4.1 section, in the manuscript). Also, the role of Gasdermin-D (GSDMD) should be detailed in the main text, and not only in the Figure legend. Otherwise, the Figure seems to contain information that is not mentioned in the text.
- Pyroptosis increases the release of mature IL-1β, as mentioned in lines 165-166. Is it, therefore, exerting a positive feedback loop effect on NLRP3 formation (as it can be extracted from Figure 2)? In this case, this reinforcing effect should be mentioned in the text and depicted in Figure 1.
- NF-κB is firstly mentioned in line 186, referred as a potential predictor of leukemia disease outcome. However, its connection with NLRP3 inflammasome is not discussed before. The authors should clarify this point and, given its importance, describe the function of NF-κB further.
- The brief introduction about myelodysplastic syndrome (MDS) in lines 191-194 would better fit in the introductory section, along with the definition of leukemia and its different subtypes.
- The order of references needs to be thoroughly revised. There are some examples of inappropriate citing and confusion between references.
- In the main text, reference 51 (line 194) appears before reference first citation of reference 44 (line 200).
- Reference 52 (line 230) is mentioned before citation number 49 (line 234), and numbers 53-55 (line 239) appear before number 50 (line 250)
- In Table 2, references 50 and 54 are probably exchanged with each other. This problem may be a result of the previous point.
- Again, in Table 2, use of reference 56 referring to MDS seems awkward. This reference should be deleted from the table or replaced by number 46.
8. The role of Th1 and Th22 lymphocytes in inflammation is mentioned in lines 228-231. Again, I believe that their role should be discussed before, in the introduction.
9. I understand that NRLP3 inflammasome’s role in CLL is opposite from its activity in other types of leukemia and, in consequence, I guess this is why it is not included in Figure 2. However, I think it should be included to complete the full picture of NLRP3 inflammasome role in hematological diseases.
10. Regarding the legend in Figure 2, it should be much shorter and more concise. As it is now, all the information in the text is mentioned, again, in the Figure Legend, which is totally unnecessary.
11. Section 4.3 is really interesting, given the importance of exploring new avenues in treating hematological diseases. Therefore, I believe that some of the inhibitors that target NLRP3 activation could be included with further detail. In general, this section merits to be further developed.
12. Legend in Figure 3 is wrong. In fact, it contains the same text that Figure 2. Authors should replace it by the correct one.
Minor points
- Use of “leukemias” in plural is weird, since the correct form should be “leukemia”. It appears in lines 62 and 86, for example (apart from the subheading of section 2). Alternative expressions like “a group of leukemias” or “distinct type of leukemias” are correct but, beginning a sentence as “The leukemias…” is not appropriate.
- In line 133, “in vivo” should be written in italics.
- The subheading “4.2.1 Myelodysplastic syndrome (MDS)” (line 190) should be in regular typing, not in italics.
Author Response
Please see teh attachement

Reviewer 2 Report
This review article is a very concise summary of the ongoing research on NLRP3 inflammasome in myeloid malignancies. Overall represents a very good overview.
Author Response
REVIEWER 2
We thank the reviewer for the very good rating and the positive review.
Round 2
Reviewer 1 Report
The review from Urwanisch et al. has been widely improved in this revised version and can be now accepted for publication in its present form.